# Structural Characterization and Glycosaminoglycan Impurities Analysis of Chondroitin Sulfate from Chinese Sturgeon

**DOI:** 10.3390/polym14235311

**Published:** 2022-12-05

**Authors:** Mei Zhao, Yong Qin, Ying Fan, Xu Wang, Haixin Yi, Xiaoyu Cui, Fuchuan Li, Wenshuang Wang

**Affiliations:** 1National Glycoengineering Research Center, Shandong Key Laboratory of Carbohydrate Chemistry and Glycobiology and State Key Laboratory of Microbial Technology, Shandong University, 72 Binhai Rd, Qingdao 266237, China; 2Qingdao Special Servicemen Recuperation Center of PLA Navy, Qingdao 266071, China; 3College of Marine Life Sciences, Ocean University of China, Qingdao 266100, China

**Keywords:** Chinese sturgeon, chondroitin sulfate, molecular weight, disaccharide composition, oligosaccharides

## Abstract

Chinese sturgeon was an endangered cartilaginous fish. The success of artificial breeding has promoted it to a food fish and it is now beginning to provide a new source of cartilage for the extraction of chondroitin sulfate (CS). However, the structural characteristics of sturgeon CS from different tissues remain to be determined in more detail. In this study, CSs from the head, backbone, and fin cartilage of Chinese sturgeon were individually purified and characterized for the first time. The molecular weights, disaccharide compositions, and oligosaccharide sulfation patterns of these CSs are significantly different. Fin CS (SFCS), rich in GlcUAα1-3GalNAc(4S), has the biggest molecular weight (26.5 kDa). In contrast, head CS (SHCS) has a molecular weight of 21.0 kDa and is rich in GlcUAα1-3GalNAc(6S). Most features of backbone CS (SBCS) are between the former two. Other glycosaminoglycan impurities in these three sturgeon-derived CSs were lower than those in other common commercial CSs. All three CSs have no effect on the activity of thrombin or Factor Xa in the presence of antithrombin III. Hence, Chinese sturgeon cartilage is a potential source for the preparation of CSs with different features for food and pharmaceutical applications.

## 1. Introduction

Chondroitin sulfate (CS) is an important member of the glycosaminoglycan (GAG) family and it consists of repeating disaccharide units containing D-glucuronic acid (GlcA) and N-acetyl-D-galactosamine (GalNAc) with various sulfation patterns [1,2]. These modifications lead to several kinds of CS/DS disaccharide units including O unit [GlcAβ1-3GalNAc], A unit [GlcAβ1-3GalNAc(4S)], C unit [GlcAβ1-3GalNAc(6S)], D unit [GlcA(2S)β1-3GalNAc(6S)], E unit [GlcAβ1-3GalNAc(4S,6S)] and T unit [GlcA(2S)β1-3GalNAc(4S,6S)], where 2S, 4S and 6S represent 2-*O*-, 4-*O*- and 6-*O*-sulfate groups, respectively. The pattern and degree of sulfation, relative content, and molecular mass vary significantly with species, age, and/or tissue of origin, and CS is involved in various biological processes, including central nervous system development [3,4,5,6,7,8], wound repair [9,10], viral attachment [11,12], growth factor signaling [13,14], morphogenesis and cytokinesis [15,16,17,18]. Based on the sulfation patterns of major disaccharide units in polysaccharide chains, CSs are classified into several subtypes, such as CS-A from bovine cartilage, CS-C from shark cartilage, CS-D from shark fin, and CS-E from squid cartilage. Due to the important biological activities of CS, it is widely used in food and pharmaceutical industries, the quality control of which is also receiving more and more attention. Therefore, it is of great significance to explore new sources for bioactive CS preparations.

The Chinese sturgeon (*Acipenser sinensis*) is an ancient cartilaginous fish native to China and is often described as a “living fossil” [19]. The population of wild Chinese sturgeon was sharply declining due to overfishing, damming, and water pollution [20,21,22]. Fortunately, recent artificial breeding programs have been greatly successful and the cultured sturgeon is now a very common food fish in China. With the increasing consumption of Chinese sturgeon as food, an abundance of discarded cartilage can be recovered and used as an excellent source for commercial CS preparation. However, the structural characteristics of Chinese sturgeon cartilage CS need to be investigated in more detail. Recently, CS purified from Chinese sturgeon cartilage was roughly characterized by gas chromatography, HPLC, the Fourier-transform infrared spectra, and ^13^C nuclear magnetic resonance (NMR) [23,24]. Moreover, several studies have shown that sturgeon CS has excellent bioactivities, such as anti-cancer [25,26], wound healing enhancement [27], and gastroprotective [28]. Nonetheless, as described before, the expression of CS is tissue/organ dependent, and thus it is necessary to characterize CSs from the different organs of Chinese sturgeon in detail. Furthermore, CS in food and healthcare products is susceptible to being contaminated by other glycosaminoglycans (GAGs). These GAG impurities, such as heparin (Hep), heparan sulfate (HS), keratan sulfate (KS), and dermatan sulfate (DS), are very similar to CS in structural characteristics and are difficult to remove, which may lead to side effects and allergic reactions. In previous studies, these GAG impurities have not been considered when studying the biological activity of sturgeon CS.

In this study, CSs were purified from the head, backbone, and fin cartilage of Chinese sturgeon, and characterized by molecular weight determination, disaccharide composition, NMR analysis, GAG impurity analysis, and internal structure analysis. In addition, the anticoagulation activity of CSs was investigated as well.

## 2. Materials and Methods

### 2.1. Materials

Chondroitinase (CSase) ABC from *Proteus vulgaris*, hyaluronidase from sheep testes, heparinase (Hepase) I, Hepase II, Hepase III, and CSase B from *Flavobacterium heparinum*, protease from *Streptomyces griseus*, CS-A from bovine trachea, CS-C from shark cartilage, sodium cyanoborohydride, D_2_O, Alcian Blue, NaH_2_PO_4_ and 2-Aminobenzamide (2-AB) were purchased from Sigma Aldrich (Shanghai, China). Keratan sulfate hydrolase (KSase) was prepared as in previous research [29]. Hep (200 IU/mg) from porcine intestinal mucosa were provided by Dongying Tiandong Pharmaceutical Co., Ltd (Dongying, China). Unsaturated disaccharides CS standards were purchased from Iduron (Manchester, UK). CS-D from shark fins was purchased from Seikagaku Corp. (Tokyo, Japan). Ethanol, acetone, NaCl, NaOH, trichloroacetic acid, ether, NH_4_HCO_3_, sodium acetate and acetic acid were purchased from China National Pharmaceutical Group Co., Ltd. (Shanghai, China). Acrylamide and bis-acrylamide were purchased from Sangon Biotech Co., Ltd. (Shanghai, China). Superdex^TM^ Peptide 10/300 GL column was obtained from GE Healthcare Life Sciences (Beijing, China). YMC-Pack PA-G column (250 × 4.6 mm ID) was purchased from YMC (Kyoto, Japan). All other reagents were of the highest quality available.

### 2.2. Preparation of Chinese Sturgeon Cartilage

A Chinese sturgeon (1 kg) was boiled in water for 15 min. Then the bone was separated from the meat and washed with water and 100% ethanol to remove the fat surrounding the bone. The cartilage was split into three parts, including head, backbone, and fins, and cut into small pieces for acetone extraction. For complete drying of the samples, the acetone-treated preparations were placed in a vacuum desiccator (Huaou Glass Co., Ltd., Yancheng, China) with P_2_O_5_ for 48 h. The dried samples were weighed and ground for CS isolation.

### 2.3. Extraction of CS from Chinese Sturgeon Cartilage

The dried cartilage powders from head, backbone, and fins were first extracted using 1 M NaOH and stirring at 45 °C for 2 h. Then the extract was collected and centrifuged at 10,000× *g* for 20 min at 4 °C. The supernatants were precipitated into four volumes of ethanol and the precipitates were used for extraction of CS as described previously with minor modifications [30]. Briefly, the crude preparations were treated with protease, followed by precipitation with 5% trichloroacetic acid to remove residual proteins and peptides. The trichloroacetic acid was removed with ether extraction and the remaining crude CS fractions were precipitated with 80% ethanol containing 5% sodium acetate at 4 °C overnight. The CS fractions were finally washed with ethanol and dried.

### 2.4. Purification of CS Preparations

The crude CS preparations from Chinese sturgeon cartilage were further loaded on anion-exchange chromatography on a DEAE Sepharose FF column (200 × 15 mm ID) pre-equilibrated with 50 mM phosphate buffer (pH 6.0) containing 0.2 M NaCl. The column was washed stepwise with 50 mM phosphate buffers (pH 6.0) containing 0.2, 0.5, and 2.0 M NaCl and the fractions obtained by elution with 2.0 M NaCl were desalted by dialysis against distilled water and lyophilized for further analysis. All purified CS samples were quantified by the carbazole reaction [31]. CS from head, backbone, and fins were named SHCS, SBCS, and SFCS, respectively.

### 2.5. Enzyme Analysis of CS Preparations

To investigate if the CS samples from sturgeon cartilages contained other GAGs impurities such as DS, KS, and Hep/HS, the purified samples (10 μg each) were treated at 37 °C for 4 h with CSase ABC (10 mIU), CSase B (10 mIU), KSase (10 mIU), and a mixture of Hepases I, II and III (4 mIU each) (Hepases mixture), respectively. The enzymatic products by CSase ABC, CSase B, and Hepases were analyzed using gel filtration chromatography on a Superdex^TM^ Peptide 10/300 GL column eluted with 0.2 M NH_4_HCO_3_ at a flow rate of 0.4 mL/min for 60 min being monitored at 232 nm by a UV detector (Shimadzu Co., Ltd., Kyoto, Japan).

To analyze the content of KS, enzymatic products (2 μg each) digested with KSase or CSase ABC were labeled with 2-AB as described before [32], and analyzed by gel filtration chromatography on a Superdex^TM^ Peptide 10/300 GL column eluted with 0.2 M NH_4_HCO_3_ at a flow rate of 0.4 mL/min using a fluorescence detector at excitation and emission wavelengths of 330 nm and 420 nm. The same method was applied to evaluate the content of KS in commercial CS-A from bovine trachea and CS-C from shark cartilage. The content of impurities is calculated by the integral area.

Based on the results from the analysis of GAG impurities, the trace DS and KS in SFCS and SBCS were removed by treatment with CSase B and KSase, and the corresponding resultants were individually precipitated with 80% ethanol containing 5% sodium acetate to collect the further purified SFCS and SBCS.

### 2.6. Analysis of Molecular Mass

The molecular weights of the purified Chinese sturgeon CS samples were compared with commercial-grade CS-A, CS-C, and CS-D by polyacrylamide gel electrophoresis (PAGE). An equal amount of each sample (5 μL at 2 mg/mL) was combined with 5 μL of 50% (*w*/*v*) sucrose in water, and the mixture was loaded into a stacking gel of 5% (total acrylamide) and fractionated with an 18.3% resolving gel. Electrophoresis was performed at 200 V for 80 min. The gel was fixed and stained with Alcian Blue in 2% (*v*/*v*) acetic acid and destained with water [33].

The molecular masses of the purified CS samples were further determined using a DAWN^®^ HELEOS^®^ II multi-angle light scattering (MALS) instrument (Wyatt Technology Co., Santa Barbara, CA, USA) combined with a Waters 515 HPLC pump with a Shodex OHPAK SB-806M HQ column (300 × 8 mm ID) and Optilab^®^ T-rEX™ refractive index (RI) detector (Wyatt Technology Co.). The molecular masses were determined by the refractive index detector using a dn/dc of 0.140 by the Wyatt ASTRA (version 5.19.1, Wyatt Technology Co., Santa Barbara, CA, USA) software. The mobile phase was 0.2 M NaNO_3_ containing 0.02% NaN_3_ at the flow rate of 0.5 mL/min.

### 2.7. Analysis of Disaccharide Composition

An aliquot (2 μg) of CS was digested with CSase ABC (10 mIU) at 37 °C for 2 h in 50 mM Tris-HCl, 50 mM NaAc, PH 8.0. The digest was labeled with 2-AB [32] glycosaminoglycan-derived oligosaccharides labeled with a fluorophore 2-aminobeand subjected to anion-exchange HPLC on a YMC-Pack PA-G column eluted with a linear gradient from 16 to 474 mM NaH_2_PO_4_ over a period of 60 min being monitored at Ex 330 nm and Em 420 nm using a fluorescence detector (Shimadzu Co., Ltd., Kyoto, Japan) [30].

### 2.8. NMR Analysis of Purified CS Samples

Purified CS samples (30 mg each) and standard CS-A (30 mg) were dissolved in 99.9% D_2_O and lyophilized twice, and were finally dissolved in 0.5 mL D_2_O in a 5 mm NMR tube. ^1^H NMR spectroscopy and ^13^C spectroscopy were recorded on Agilent DD2 600 (Agilent Technologies Inc., Santa Barbara, CA, USA) operating at a proton frequency of 600 MHz.

### 2.9. Analysis of Size-Defined Oligosaccharides

To compare the internal structures of CS chains from the cartilage of different parts of Chinese sturgeon, 10 μg of each purified CS sample was partially digested at 37 °C for 24 h with 7.5 U of hyaluronidase from sheep testes. The digests were labeled with 2-AB and fractionated on a Superdex^TM^ Peptide 10/300 GL column eluted with 0.2 M NH_4_HCO_3_ at a flow rate of 0.4 mL/min for 60 min with fluorescent detection. The 2-AB-labeled oligosaccharide fractions were collected, freeze-dried, and analyzed by anion-exchange HPLC on a YMC-Pack PA-G column eluted with a linear gradient of NaH_2_PO_4_ at a flow rate of 1 mL/min at room temperature.

### 2.10. Analysis of Anticoagulant Activity

To investigate the anticoagulation activity of purified CS samples, different concentrations (0–5 mg/mL each) of CS samples were successively incubated with antithrombin III (ATIII) and thrombin (factor IIa, FIIa) or factor Xa (FXa). The residual activity of FIIa or FXa was analyzed by using an anticoagulant kit (Aglyco kit, Beijing Adhoc International Technologies Co., Ltd., Beijing, China). Hep was set as a reference.

## 3. Results

### 3.1. Preparation of CS from Chinese Sturgeon Cartilage

As performed in the large-scale production of CS, we extracted CS from Chinese sturgeon cartilage with alkali first in order to save the relatively expensive protease used in the following step. The alkali extracts from the head, backbone and fin cartilages of Chinese sturgeon were exhaustively digested by protease, and crude CSs individually recovered with ethanol precipitation. All three CS samples were further purified using anion-exchange chromatography on a DEAE-Sepharose FF column as described under “Section 2”. After desalting, the purified CS preparations were quantified using the carbazole reaction. As shown in Table 1, the weight of the head, backbone, and fin cartilages account for 56.7%, 21.6%, and 21.7% of total cartilages, respectively, and CS preparations purified from them have relatively high yields from 25.63 to 30.74% with high purities from 96.0 to 98.6%.

### 3.2. Enzyme Analysis of CS Preparations

To investigate whether CS from Chinese sturgeon cartilage contains DS, KS, Hep/HS or not, the SHCS, SBCS, and SFCS (10 μg) were treated with CSase ABC, CSase B, KSase, or Hepases mixture, respectively. As shown in Figure 1, while treated with CSase ABC, more than 95% of CS preparations were digested to monosulfated disaccharides. No significant oligosaccharide peak was observed in CS preparations when treated with a Hepases mixture or CSase B, while only one small peak of monosulfated disaccharides was detected in the digest of fin cartilage CS. These results indicate that CS chains from Chinese sturgeon cartilage contain little to no heparinoid or DS, and the trace amount of DS in the fin CS may be contaminants from residual skin or meat. When CS preparations were treated with KSase, one single small peak of monosulfated disaccharides was detected in fin and backbone cartilage which indicates CS chains from fin and backbone cartilage contain trace KS (<2% of all GAGs). In addition, the KS content of commercial CS-A and CS-C was also evaluated. As shown in Figure 1D, the result showed that KS was clearly present in commercial CS-A (approximately 10.8%), and trace KS was detected in commercial CS-C (approximately 2%). This result suggests that sturgeon CS is a good source of CS due to its minimal GAG impurities. The subsequent tests were performed after all GAGs impurities were completely removed by degrading with GAG lyases.

### 3.3. Analysis of the Molecular Mass

The molecular sizes of the CS preparations from head, backbone, and fin cartilages were compared to commercial-grade CS preparations by PAGE. As shown in Figure 2, the molecular sizes of all CS samples are smaller than CS-C from shark cartilage and CS-D from shark fins, but slightly larger than CS-A from bovine trachea. In addition, the molecular size of fin CS is significantly bigger than that of head and backbone CSs. Their molecular masses were further determined using GPC-RI-MALS (Figure 3). The average molecular masses of head, backbone and fin CSs are 21.0, 20.2, and 26.5 kDa, respectively (Table 2), which are consistent with the results from PAGE. To present the molecular weight distribution of these CSs, the polydispersion index (PDI) was calculated by Mw/Mn, and the results showed that the PDI values of SHCS, SBCS, and SFCS were 1.07, 1.07, and 1.05, respectively, indicating that the molecular weight distribution of CS from Chinese sturgeon cartilage is quite narrow.

### 3.4. Analysis of the Disaccharide Composition

To identify the disaccharide composition of CSs from Chinese sturgeon cartilages, three CS preparations were digested with CSase ABC, labeled by 2-AB, and then analyzed by anion-exchange HPLC, as described under “Section 2”. As shown in Figure 4 and Table 3, all CS preparations showed the presence of Δ^4,5^HexUAα1–3GalNAc (ΔO unit), in which Δ^4,5^HexUA represents unsaturated uronic acid, Δ^4,5^HexUAα1–3GalNAc(6S) (ΔC unit) and Δ^4,5^HexUAα1–3GalNAc(4S) (ΔA unit) but in varying proportions. Digestions by CSase B, which specifically cleave DS but not CS, showed that the CS preparations from Chinese sturgeon cartilage contained no obvious DS (data not shown), which suggested that the ΔO, ΔC, and ΔA units released by CSase ABC digestion are derived from CS disaccharide GlcUAα1–3GalNAc (O unit), GlcUAα1–3GalNAc(6S) (C unit) and GlcUAα1–3GalNAc(4S) (A unit), respectively, in the CS chains. Notably, the proportions of C and A units as main components are quite different among the three preparations. Head and fin cartilage CSs predominantly contain C (56.56%) and A units (54.93%), respectively, suggesting that these CSs have very different internal structures.

### 3.5. NMR Spectroscopy

The ^1^H and ^13^C NMR spectroscopy of CS preparations from the backbone, fins, head, and standard CS-A were recorded, and their chemical shifts were assigned based on previous studies [34,35,36]. Judging from the similar intact ^1^H and ^13^C NMR spectroscopy, the three CS preparations possibly have extremely analogous structures with each other and with standard CS-A. In ^1^H NMR spectroscopy (Figure 5A), the signals at 2.02 ppm were attributed to the methyl protons of GalNAc, and the signals at 4.53 ppm and 4.48 ppm were assigned to H-1 of GalNAc and GlcA, respectively. The proton signals of carbon rings were distributed in 4.25 ppm to 3.26 ppm. Notably, the resonances at 4.10 ppm were characteristic of H-6 of GalNAc(6S) residues which proves the presence of sulfate substitution at the C-6 position of GalNAc. No signals were recorded in the 5.0–6.0 ppm region indicating that no alpha hexopyranose residues were present, such as alpha-iduronic acid or alpha-glucosamine. In ^13^C NMR spectroscopy (Figure 5B), the resonances at 175.0 and 22.5 ppm were ascribed to the signals of carbonyl and acetamido methyl carbons. The signals in the 104.5–103.0 ppm and 101.9–100.1 ppm regions were assigned to the C-1 of GlcA and GalNAc, respectively. The signals at 76.5 ppm were considered as the resonances of C-4 of GalNAc(4S), while the signals at 67.2 ppm were assigned to the C-6 of GalNAc(6S). These results revealed that the three CS preparations all contained GlcA, GalNAc, GalNAc(6S), and GalNAc(4S) residues which were consistent with the analysis of the disaccharide composition.

### 3.6. Analysis of Internal Structure

To compare the structures of CS chains from different organ cartilages of Chinese sturgeon, three CS preparations were partially digested by hyaluronidase from sheep testes, labeled by 2-AB, and fractionated using gel filtration chromatography. The collected size-defined oligosaccharide fractions were further analyzed by anion-exchange HPLC as described under “Section 2”. All three CS samples were predominately digested into tetra-, hexa-, octa-, and decasaccharides. The proportions of oligosaccharides in the digests of head and backbone CSs were similar but the proportions of octa- and decasaccharides in the digest of fin CS were significantly lower than those in the former two (Figure 6 and Table 4).

The size-defined oligosaccharide fractions were further analyzed using anion-exchange HPLC. As shown in Figure 7, all three tetrasaccharide fractions prepared from the head, backbone, and fin CSs were subfractionated into three major peaks between 33.7 and 37.7 min but the area ratio of the peaks is different. Compared to the tetrasaccharide fractions prepared from head CS, the ratio of subfraction eluted by the high concentration of salt significantly increases in tetrasaccharide fractions from the backbone and fin CS, which indicates that the tetrasaccharide domains are more negatively charged in the CS chains from the backbone and in particular fins. Hexasaccharide fractions from three CS preparations were mostly resolved into two clusters of peaks eluted from 37.5 to 43.5 min. The first group with a lower negative charge contains three partially overlapped peaks and the second group with a higher negative charge has two well-separated peaks. The elution patterns of hexasaccharides also suggest that the negative charge of hexasaccharide domains is highest in fin CS chains followed successively by backbone and head CSs (Figure 8). Similar results were observed in octa- and decasaccharide fractions prepared from three CS preparations too (data not shown). These results demonstrate that the internal structures of CS chains from the cartilages of the head, backbone, and fins are significantly different.

### 3.7. Effect of Three CS Preparations to the FXa or FIIa in the Presence of ATIII

To investigate the anticoagulant activity of CSs from Chinese sturgeon cartilages, three CS preparations were incubated with FXa or FIIa in the presence of ATIII, and the residue activity of FXa or FIIa was recorded at OD405. However, no significant changes in whether FXa or FIIa OD405 were detected which indicated there was no inhibitory effect on the activity of FXa or FIIa for all three CS preparations (0–5 mg/mL) (Figure 9). Under the same conditions, Hep showed a strong ability to inhibit the activity of FXa or FIIa in the presence of ATIII even at very low concentrations (0–0.8 μg/mL). This result is evidence that all three CS preparations are much less potent in the ATIII-mediated inhibition of FIIa and FXa when compared with heparin.

## 4. Discussion

The various biological functions of CS are attributed to their structural diversity. In biosynthesis, either after or during the polymerization of the basic structure, which is composed of repeating disaccharide units of GlcUA-GalNAc, CS chains are modified by specific sulfotransferases as well as glucuronyl C5 epimerase to yield the diverse structures in cell/tissue-specific manner [37,38,39]. These unique structures are introduced into CS/DS chains in the specific tissues resulting in differing responses to CS/DS-binding proteins such as heparin-binding growth factors and cytokines [1]. In this study, CS chains purified from the head, backbone, and fin cartilages of Chinese sturgeon show significant differences in molecular weights, disaccharide compositions, and internal structures.

CS has been widely used in the treatment of osteoarthritis. It is undeniable that the high purity and high quality of CS are the basis of its application and safety. In the current process of CS separation and purification, there are still some potential contaminants such as other polysaccharides, proteins, and some residual organic compounds. Among them, GAG polysaccharides have similar structures and are usually difficult to distinguish. Therefore, we analyzed the three CS preparations with different GAGs-lyase enzymes. The results showed that the three CS preparations contain little to no Hep/HS, DS, and KS (1–2.5%), which is much lower than commercial CS preparations from shark and bovine cartilages manufactured by pharmaceutical companies of oral administration (1–16%), and less quality controlled products (2–47%) [40]. It is indicated that sturgeon cartilage is a new source for the preparation of CS.

Commercial CS is predominately purified from bovine trachea, porcine nasal septa, chicken keel, and shark cartilage. The CS characteristics such as molecular weight and sulfation pattern vary among the different organisms. Typically, the CS molecular weight from marine cartilaginous fishes, such as sharks and raja is 50–70 kDa but the Chinese sturgeon CS has a much lower molecular mass (20–27 kDa), which is more similar to CSs from terrestrial animals like avian, porcine, and bovine [41,42]. Two previous studies reported that the molecular weights of Chinese sturgeon cartilage CS were 13.4 kDa [24] and 43 kDa [23], respectively. Notably, the latter may be an overvalued result due to the analysis method using a TSK G6000 PWXL column with a separation range of 500–50,000 kDa for dextran which is outside the range for CS from Chinese sturgeon [23].

Disaccharide analysis showed that CSs from mammals and avians, such as porcine, bovine, and chicken, mainly consist of monosulfated disaccharide A and C units with a low proportion of O unit, where the A unit is the main component and the ratio of A to C unit is from 1.5 to 7.0 [42], which are usually classified into the subtype of CS-A. In contrast, CS from shark cartilage is the only commercially available CS with a high proportion of C units, where the ratio of C to A unit is from 1.4 to 2.2 [42], which thus is designated as CS-C. In this study, we found that the CS from Chinese sturgeon cartilage was structurally similar and possessed subtle differences in fine structure. The ratio of A to C units in CS from Chinese sturgeon cartilage is highly organ-dependent and head CS with 56.56% of C units and fin CS with 54.93% of A units can be classified into the subtypes CS-C and CS-A, respectively. Thus, Chinese sturgeon cartilage will be a good source for the preparation of different CS isoforms for various applications.

Commercial CS preparations, even high sulfated CS-D from shark fin and CS-E from squid cartilage [43,44], usually have an extremely lower anticoagulant activity than that of Hep [45]. Exceptionally, fucosylated CS from sea cucumber have remarkably high anticoagulant activity [46] and can become a promising oral anticoagulant antithrombotic agent [47]. In this study, the inhibition of three CS preparations to FXa or FIIa was investigated in the presence of ATIII. The results showed that three CS preparations had no effect on the inhibition of thrombin and factor Xa by ATIII. However, some studies have confirmed that CS-A or CS-C could prolong prothrombin time, thrombin time and activated partial thromboplastin time [48,49], the anticoagulation mechanism of CS-A/CS-C is thought to function independently of AT III and the anticoagulation may act by directly preventing thrombin generation [50]. Our results also proved that CS preparations mainly containing C unit or A unit have no effect on factor Xa/IIa inhibition. Interestingly, Gui M. et al. reported that CS-C from sturgeon backbone (59.6% of C unit and 37.8% of A unit) could inhibit FXa mediated by ATIII [51], and the difference in experimental results between Gui and us needs further analysis and discussion.

The various bioactivities of CS are determined by some oligosaccharide domains with specific sulfation patterns, which are usually longer than disaccharides. Therefore, investigating the features of oligosaccharides longer than disaccharides is a key task for elucidating the structural and functional relationships of CS chains. In the present study, a series of size-defined oligosaccharides derived from three CS preparations were analyzed by anion-exchange HPLC, and the elution profiles of oligosaccharides suggested that the internal structures of CS chains from the different cartilage tissues of Chinese sturgeon were significantly different, which further indicated that the bioactivities of three CS preparations should be different too.

## Figures and Tables

**Figure 1 polymers-14-05311-f001:**
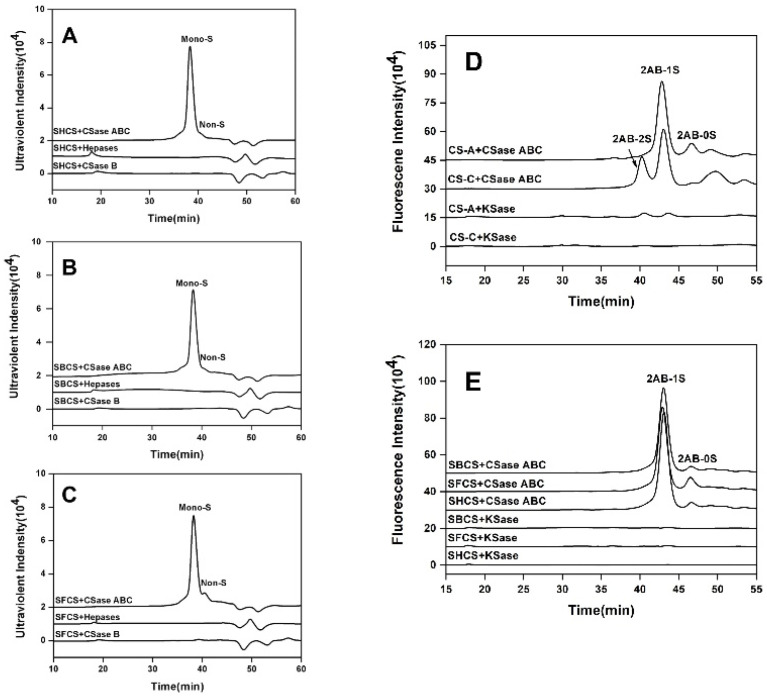
Analysis of GAG-lyase-treated CS preparations by gel filtration chromatography. SHCS (**A**), SBCS (**B**), and SFCS (**C**) were treated with CSase ABC, Hepases, and CSase B, respectively, and the resultants were analyzed by gel filtration with a UV detector. Mono-S, the monosulfated disaccharides; Non-S, non-sulfated disaccharides. To determine the content of KS impurity in commercial CS-A, CS-C (**D**), or three CS preparations (**E**), each CS sample was digested with KSase and CSase ABC, respectively, and the product was labeled with 2-AB followed by analysis by gel filtration with a fluorescence detector. 2AB-2S, 2AB-labeled disulfated disaccharides; 2AB-1S, 2AB-labeled monosulfated disaccharides; 2AB-0S, 2AB-labeled non-sulfated disaccharides.

**Figure 2 polymers-14-05311-f002:**
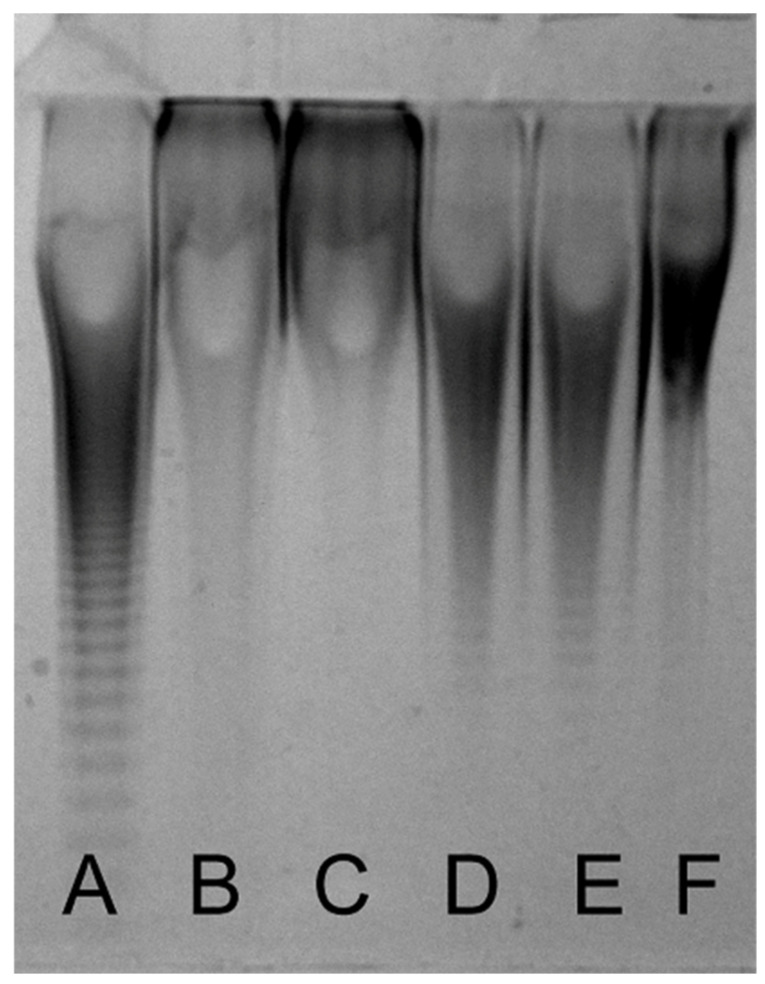
Analysis of Chinese sturgeon cartilage CSs by PAGE. The CS preparations from head, backbone, and fin cartilages of Chinese sturgeon were analyzed and compared with commercial CSs by PAGE using 18.3% polyacrylamide gels followed by staining with Alcian Blue, as described under “Section 2”. Lane A: CS-A; Lane B: CS-C; Lane C: CS-D; Lane D: SHCS; Lane E: SBCS; Lane F: SFCS.

**Figure 3 polymers-14-05311-f003:**
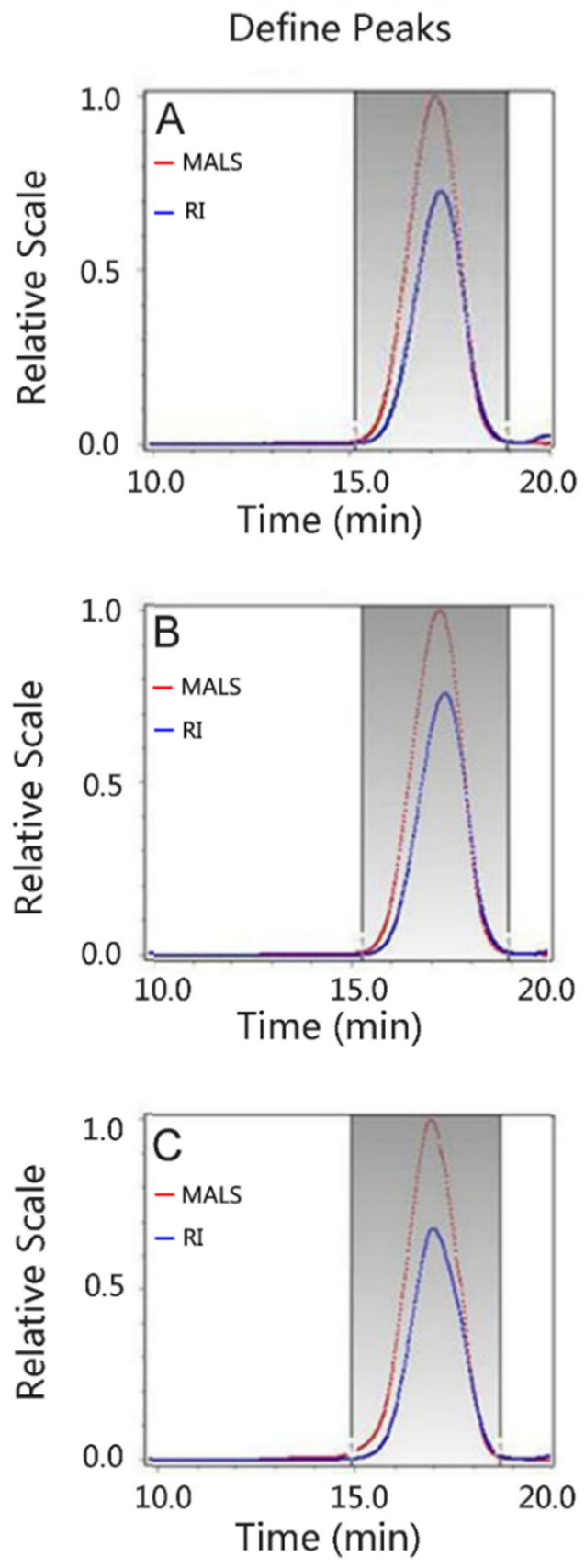
Molecular mass analysis by MALS. The molecular mass of SHCS (**A**), SBCS (**B**), or SFCS (**C**) was measured by size exclusion chromatography with a multi-angle light scattering (MALS) detector and a refractive index (RI) detector, as described under “Section 2”.

**Figure 4 polymers-14-05311-f004:**
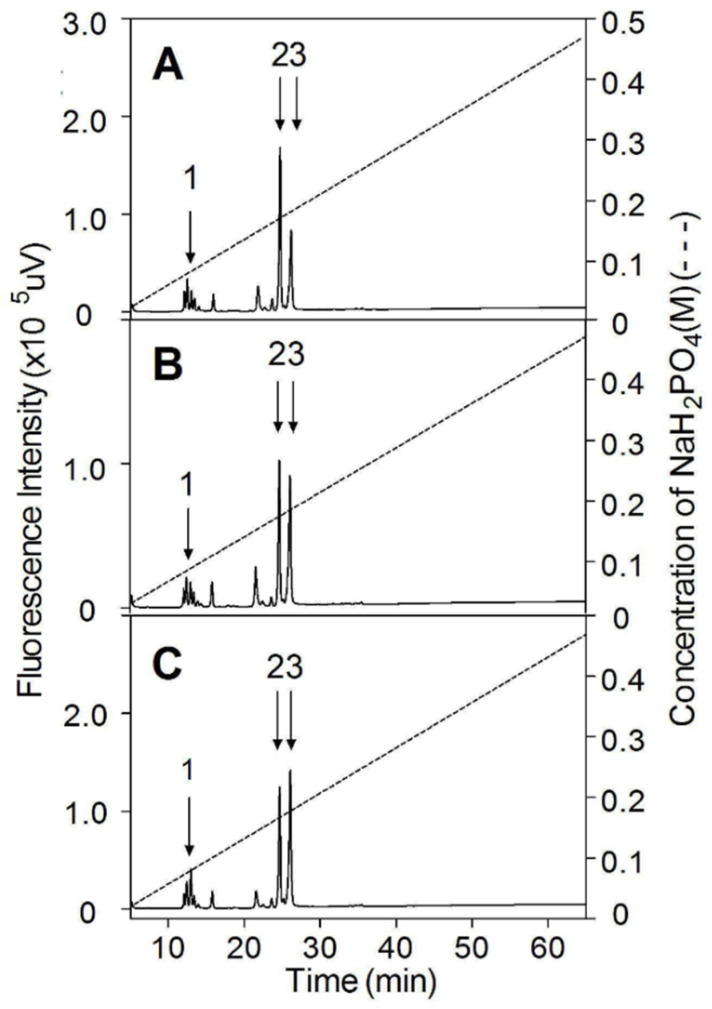
Disaccharide composition analysis by anion-exchange HPLC. CS (2 μg) from the cartilage of head (**A**), backbone (**B**), or fins (**C**) were digested with CSase ABC, labeled by 2-AB, and analyzed by an anion-exchange HPLC column with a fluorescence detector. All samples were analyzed by HPLC on a YMC-Pack PA-G column using a NaH_2_PO_4_ gradient (indicated by the dashed line). The elution positions of authentic 2-AB-labeled unsaturated disaccharides are indicated by arrows. 1, Δ^4,5^HexUAα1-3GalNAc; 2, Δ^4,5^HexUAα1-3GalNAc(6S); 3, Δ^4,5^HexUAα1-3GalNAc(4S).

**Figure 5 polymers-14-05311-f005:**
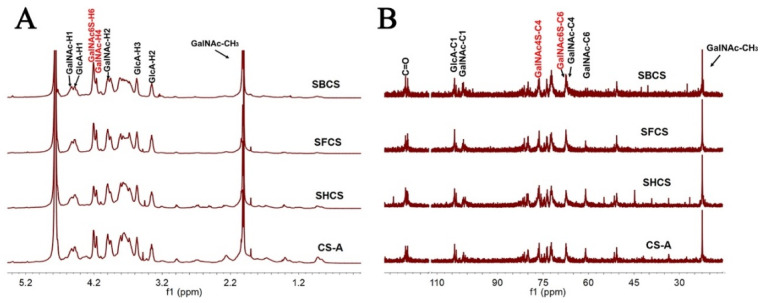
NMR analysis of three CSs and CS-A. (**A**) ^1^H NMR spectra. (**B**) ^13^C NMR spectra.

**Figure 6 polymers-14-05311-f006:**
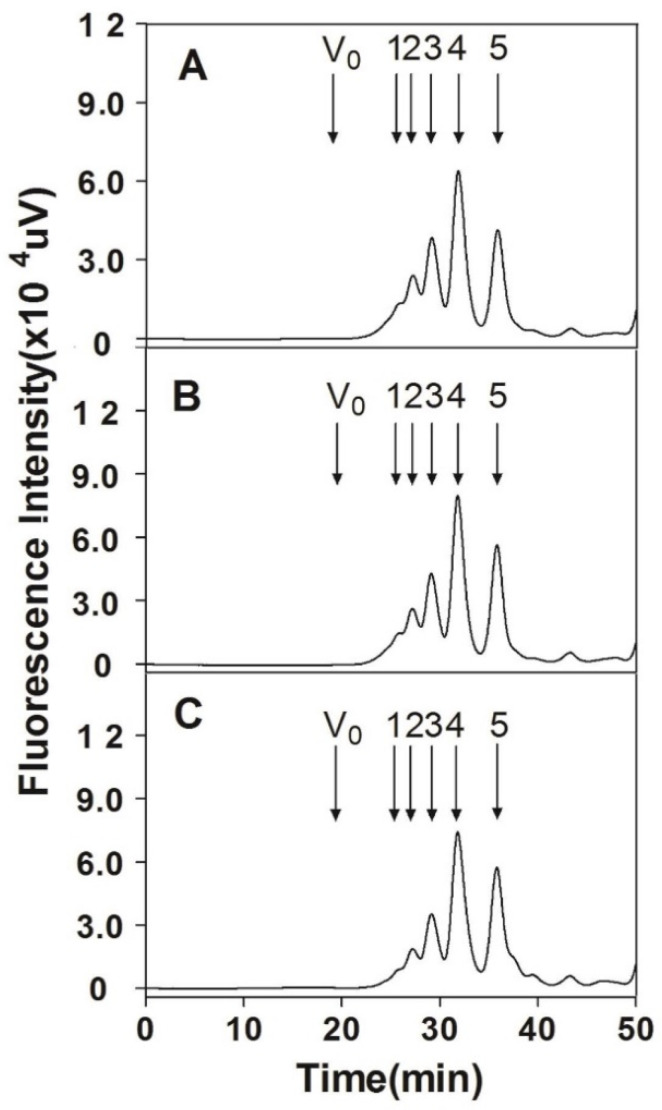
Preparation of size-defined oligosaccharides. SHCS (**A**), SBCS (**B**), or SFCS (**C**) was digested with hyaluronidase, and after 2-AB labeling, the digest was fractionated on a Superdex^TM^ Peptide 10/300GL column as described under “Section 2”. Vo, void volume. The elution positions of 2AB-labeled authentic unsaturated CS-derived standard oligosaccharides determined are indicated by arrows as follows: 1, CS-dodecasaccharides; 2, CS-decasaccharides; 3, CS-octasaccharides; 4, CS-hexasaccharides; 5, CS-tetrasaccharides.

**Figure 7 polymers-14-05311-f007:**
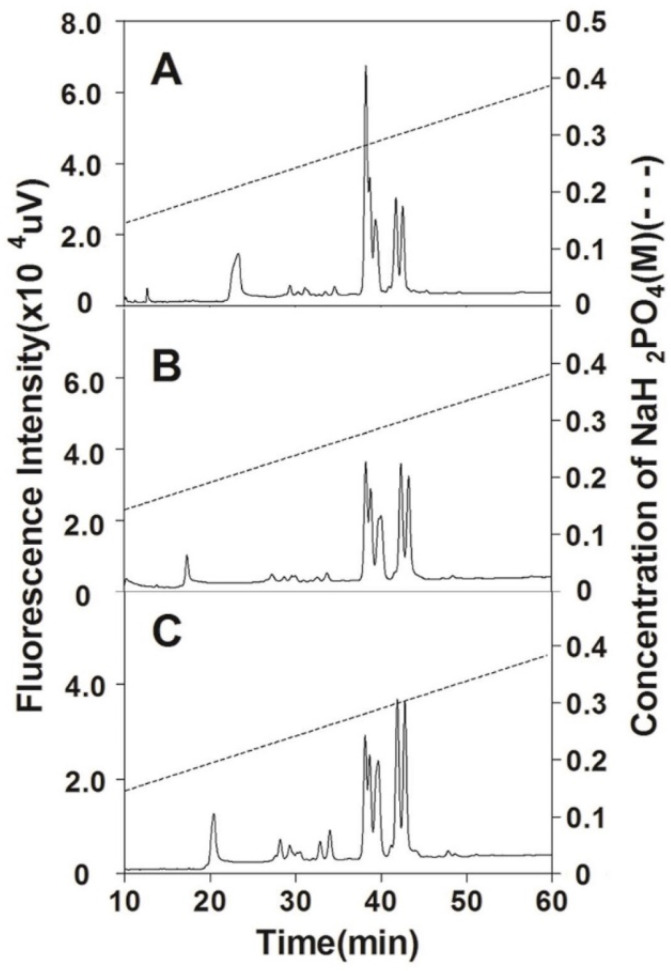
Analysis of tetrasaccharide fractions by anion-exchange HPLC. The tetrasaccharide fractions prepared from SHCS (**A**), SBCS (**B**), and SFCS (**C**) were analyzed by anion-exchange HPLC on a YMC-Pack PA-G column using a NaH_2_PO_4_ gradient (indicated by the dashed line).

**Figure 8 polymers-14-05311-f008:**
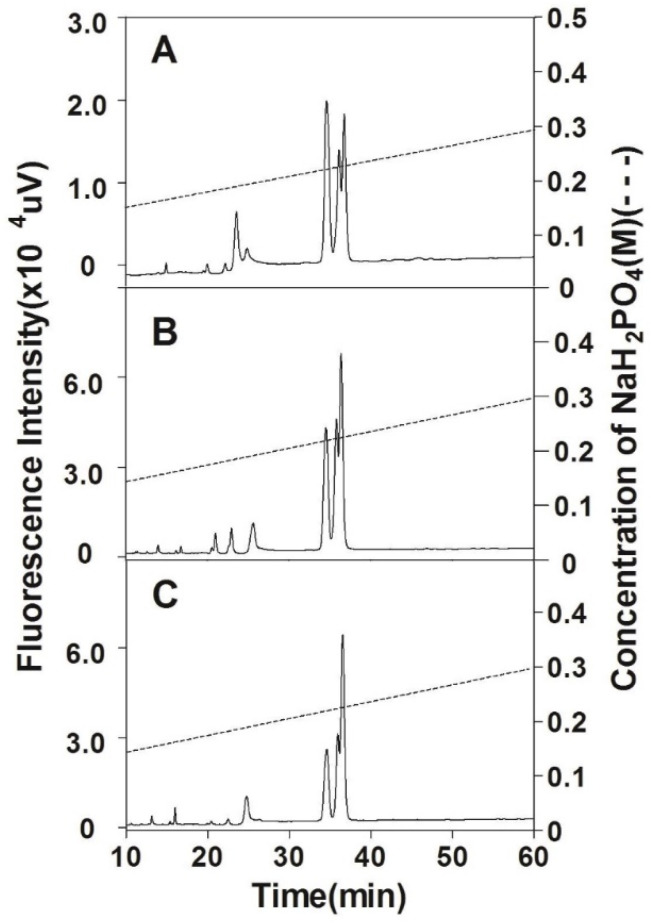
Analysis of hexasaccharide fractions by anion-exchange HPLC. The hexasaccharide fractions prepared from SHCS (**A**), SBCS (**B**), and SFCS (**C**) were analyzed as described in the caption of Figure 5. All samples were analyzed by HPLC on a YMC-Pack PA-G column using a NaH_2_PO_4_ gradient (indicated by the dashed line).

**Figure 9 polymers-14-05311-f009:**
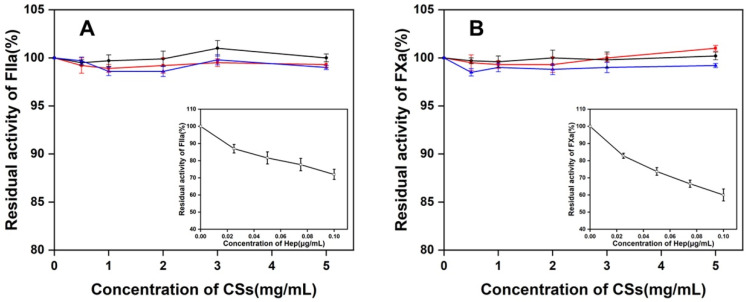
The effect of CSs from sturgeon cartilage to the effect of residual FIIa (**A**) or FXa (**B**) activity in the presence of ATIII. The standard Hep (□) was set as a reference (Insert). (●) SHCS; (■), SBCS; (▲) SFCS.

**Table 1 polymers-14-05311-t001:** The Purity and Yield of CS Preparations from Chinese Sturgeon Cartilage.

CS Samples	SHCS (%)	SBCS (%)	SFCS (%)	Total (%)
Dry weight of cartilage (g)	4.87 (56.7)	1.86 (21.6)	1.87 (21.7)	8.60 (100)
Purified CS (g)	1.30 (53.3)	0.58 (23.8)	0.56 (22.9)	2.44 (100)
Purity (%)	96.0	98.59	96.63	-
Yield (%)	25.63	30.74	28.94	-

**Table 2 polymers-14-05311-t002:** The Molecular Weight of CS Preparations from Chinese Sturgeon Cartilage. The values in the brackets correspond to the uncertainties.

CS Samples	Mn ^a^	Mw ^b^	Mz ^c^	Mw/Mn ^d^
		kDa		
SHCS	19.6 (0.4%) ^e^	21.0 (0.4%)	22.7 (0.4%)	1.07
SBCS	18.4 (0.4%)	20.2 (0.4%)	21.8 (0.4%)	1.07
SFCS	25.2 (0.4%)	26.5 (0.4%)	28.4 (0.4%)	1.05

^a^ Mn, number–average molecular weight. ^b^ Mw, weight–average molecular weight. ^c^ Mz, Z-average molecular weight. ^d^ The value of Mw/Mn means polydispersity index. ^e^ The values in the brackets correspond to the uncertainties.

**Table 3 polymers-14-05311-t003:** The Disaccharide Compositions of CS Preparations from Chinese Sturgeon Cartilage.

CS Samples	SHCS	SBCS	SFCS
		mol %	
ΔO unit	7.20	5.60	5.37
ΔC unit	56.56	44.08	39.70
ΔA unit	36.24	50.32	54.93
Total	100	100	100
S/unit	0.93	0.94	0.95

ΔO unit, Δ^4,5^HexUAα1-3GalNAc; ΔC unit, Δ^4,5^HexUAα1-3GalNAc(6S); ΔA unit, Δ^4,5^HexUAα1-3GalNAc(4S). S/unit, a molar ratio of sulfate to disaccharide.

**Table 4 polymers-14-05311-t004:** The Ratio of Oligosaccharides in the Hyaluronidase Digests of CS Preparations from Chinese Sturgeon Cartilage.

Oligosaccharides	SHCS	SBCS	SFCS
	mol %		
Tetrasccharide	26.55	28.41	32.77
Hexasaccharider	37.88	38.59	39.16
Octasaccharide	22.48	20.70	17.71
Decasaccharide	13.07	12.30	9.85
Total	100	100	100

## Data Availability

All relevant data generated during this study or analyzed in this published article are available from the corresponding author upon reasonable request.

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
