# Peer review of "Structural Characterization and Glycosaminoglycan Impurities Analysis of Chondroitin Sulfate from Chinese Sturgeon"

_polymers, 2022, doi:10.3390/polym14235311_

Round 1

Reviewer 1 Report

The manuscript “Structural Characterization and glycosaminoglycan impurities analysis of Chondroitin Sulfate from Chinese Sturgeon” is well structured and the study well planned. However, the presentation of the results needs to be improved and several points clarified. A few specific comments and suggestions of changes are included.

Abstract

Page 1, lines 25 and 26 – The results obtained in this study didn’t allow concluding chondroitin sulfate extracted from Chinese sturgeon cartilage may have pharmaceutical applications. On the other hand, the food applications of Chinese CS were not studied in this work.

Introduction

Page 1, line 32 – I suppose it is “...family and it consists of…”

Page 1, line 39 – I think it is “…and CS is involved in…” Please check.

Page 2, line 51 – I suggest replacing “successes” with “succeeded”.

Page 2, line 62 – I suggest “…susceptible to be contaminated…”

Page 2, line 65 – I suggest including “to” after “lead”.

Page 2, line 71 – I suggest “was” instead of “were”.

Materials and Methods

Page 2, line 76 – Streptomyces griseus in italics.

Page 3, line 119 – It is “To analyze”.

Page 4, line 164 – I suggest deleting “And”.

Results

Page 4, line 177 – The percentages are not shown in Table 1 and it is not explained how they were calculated.

Page 5, line 187 – Is it “non-sulfated disaccharides” instead of “monosulfated disaccharides”? Please check.

Page 5, line 194 – I suppose the results of CS-A and CS-C treatment with KSase are shown in Figure 1D. Please check.

Page 5, lines 197 and 198 – The removal of GAGs impurities are not described in the Materials and Methods.

Page 5, lines 200-203 – The legend of Figure 1 need to be improved. Where is Figure 1E?

Page 5, line 206 – Is it “polyacrylamide gel electrophoresis”?

Page 5, lines 207-209 – The Figure 2 doesn’t provide any information on the molecular weight of CS samples.

Page 5, lines 211 and 212 – The values of molecular masses (21.0, 20.2, and 26.5 kDa) are not presented in Table 2. Please check.

Page 6, lines 212, 213 and 216 – Please see my comment on line 206.

Page 6, lines 215-218 – Please improve the legend of Figure 2 and as well as this figure.

Page 6, lines 223 and 224 – Please explain the different designations of units O, A and C.

Page 7, Table 2 – Please comment the results shown in Table 2.

Page 8, Table 3 – According to Figure 4 B, it seems that C unit had higher percentage than that reported in Table 3. Please check.

Page 8, line 250 – I suppose it is “…based on previous…”

Page 10, line 301 – According to Figure 8, it seems the order is “... by backbone and head…” Please check.

Page 12, line 322 – As shown in Figure 9, it seems CS compounds had no effect on the activity of thrombin of Factor Xa in the presence of ATIII. Please check this sentence.

Discussion

Page 12, line 329 – I suggest “various” instead of “varying”.

Page 12, line 348 – I suggest replacing “source to” with “source for the”.

Page 13, lines 352, 353, and 355 – It is “kDa”.

Page 13, line 379 – There is no reference 42.

Page 13, lines 381 and 382 – I suppose it is “containing C unit or A unit…” Please check. Meng et al. is not in the References.

Page 13, line 384 – Please revise this sentence for its improvement.

References

Page 15, lines 472, 495, and 498 – The references 26, 38, and 39 are not quoted in the manuscript. Please check.

Page 15, line 486 – It is gigas.

Page 15, line 500 – It is macrocephalus.

Reviewer 2 Report

From my point of view and once the manuscript has been read in detail. I think that in the current appearance the article is correct to be published.

There are, however, a  concerns still left that need a thorough and careful attention:

*update referências.

*update fonts and standardize text font in graphics

att.

Author Response

Response to Reviewer 2 Comments

Comments and Suggestions for Authors

From my point of view and once the manuscript has been read in detail. I think that in the current appearance the article is correct to be published.

There are, however, a concerns still left that need a thorough and careful attention: *update referências.

*update fonts and standardize text font in graphics

Our response

Thanks, we have revised the manuscript and update references according to your suggestions.

Reviewer 3 Report

This manuscript describes samples of chondroitin sulfate extracted from cartilage of the Chinese sturgeon, which is farmed for food use. Chondroitin sulfate (CS) could be a useful by-product of  this industry. CS was prepared by an established method from head, backbone and fins of the fish, and these samples were analysed in terms of structural characterisation, content of other glycosaminogycans, and anticoagulant activity.

The three samples were not identical, particularly in their disaccharide analysis, with the C unit predominant in head CS but the A unit predominant in backbone and fin CS. They had no measurable activity in an antithrombin-mediated anticoagulant assay.

The paper is well presented and clearly written, apart from a few points at which minor revision would be appropriate.

1.       Lines 134-139: Molecular weights were measured by size exclusion chromatography using the MALS technique. Could the authors please record the value they used for the parameter dn/dc? This is the coefficient that governs the relationship between refractive index and concentration of the CS sample, and it is important because without this value the measurement cannot be reproduced in another laboratory.

2.       Lines 258-260: This is not correct: the 5.0 to 6.0 ppm region in proton NMR of hexopyranoses contains the alpha, not the beta, anomeric signals. Suggested revision: ‘No signals were recorded in the 5.0-.0 ppm region indicating that no alpha hexopyranose residues were present, such as alpha-iduronic acid or alpha-glucosamine.’

3.       Lines 321-323: The assay described does not measure the generation of thrombin or factor Xa. Suggested revision: ‘This result is evidence that the CS compounds are much less potent in the antithrombin-mediated inhibition of thrombin and factor Xa when compared with heparin.’

4.       Lines 376-377: suggested revision ‘The results showed that the three CS preparations had no effect on the inhibition of thrombin and factor Xa by ATIII.’
